# Sand Discharge Simulation and Flow Path Optimization of a Particle Separator

**DOI:** 10.3390/e25010147

**Published:** 2023-01-11

**Authors:** Zhou Du, Yulin Ma, Quanyong Xu, Feng Wu

**Affiliations:** 1School of Mechanics and Engineering, Liaoning Technical University, Fuxin 123000, China; 2Institute for Aero Engine, Tsinghua University, Beijing 100084, China; 3AECC Sichuan Gas Turbine Establishment, Mianyang 621000, China

**Keywords:** particle separator, structure optimization, boundary conditions, separation efficiency, numerical simulation

## Abstract

A numerical simulation method is used to optimize the removal of sand from a helicopter engine particle separator. First, the classic configuration of a particle separator based on the literature is simulated using two boundary conditions. The results show that the boundary conditions for the total pressure inlet and mass flow outlet are much more closely aligned with the experimental environment. By modifying the material at the front of the shroud, the separation efficiencies of coarse Arizona road dust (AC-Coarse) and MIL-E-5007C (C-Spec) can be improved to 93.3% and 97.6%, respectively. Configuration modifications of the particle separator with dual protection can increase the separation efficiencies of AC-Coarse and C-Spec to 91.7% and 97.7%.

## 1. Introduction

When a helicopter takes off and lands or hovers at a low altitude in the desert, snow, grassland, sea, or other harsh field conditions, the collisions of large sand particles with the high-speed rotating compressor blades may cause the blades to crack or even break. Sand particles can cause great damage to the working efficiency and service life of the engine. Smaller-sized particles can enter the engine flow channel because they are difficult to separate. These smaller-sized particles will pass through the combustion chamber with the airflow after being heated and melted, thereby blocking the cooling passage of the turbine, which causes changes in the cooling characteristics of the turbine blade. Therefore, the effectiveness of air intake protection is closely related to flight safety.

The particles taken in by helicopters are mainly sand dust and are primarily composed of SiO_2_. In nature, sand particles have a wide size range from 1 μm to 1 mm. Various forms of helicopter air intake protection devices have been proposed to protect helicopter engines and increase their service lives, such as inlet barrier filters (IBF), vortex tube separator (VTS), and integer particle separators (IPS). In 1969, the JFTD-12-4A turboshaft engine in the CH-54 helicopter was replaced due to sand abrasion after flying in Southeast Asia for less than 60 h. The average replacement life of the type of engine was only about 80 h. After a particle separation device was installed, its lifespan increased to 800 h, a tenfold increase. Among the particle separators, IPS has become the most popular and widely used air intake protection scheme due to its simple structure, small total pressure loss, and low maintenance cost. It is composed of vortex blades, reverse vortex blades, inner rings, blowers, sweeping volute, etc. The fixed vortex blades cause the air to rotate. The rotating air uses the centrifugal force of inertia to separate and force sand, gravel, dust, and other foreign objects into the outer channel of the inner ring, where they are blown out of the engine by the blower, and clean air is drawn into the engine through the inner ring.

With respect to structural optimization of the particle separator, much research has been carried out using finite volume methods, where structure optimization calculations are used to improve the separation efficiency and reduce the total pressure loss of the particle separator. Breitman [1] studied the shape of the splitter and found that the shape of the splitter has a greater impact on the performance of the IPS. When designing the splitter, local flow conditions should be considered when selecting the appropriate shape and size of the splitter. Mann [2] proposed a new particle separator configuration and analyzed the two contradictions in the particle separator design. The scavenge region requires a larger channel area to capture more pollutants, but at the same time, it also needs a smaller channel area to reduce the volume of air passing through. The core region needs a larger channel area to have more air volume, and it also needs a smaller channel area to reduce the entry of pollutants. Kuang [3] compared the different positions of the bulge and the position of the splitter tip and found that the overall structure of the particle separator must be considered comprehensive. If only the optimization and configuration of a single position are considered, it is difficult to meet these requirements. Taslim [4] modified the inlet of the existing particle separator, explored the influence of the engine inlet angle on the separation efficiency and total pressure, and studied the influence of the size, distribution, particle density, and particle shape of the engine inlet on the removal efficiency.

Many studies have been performed with respect to the most important factors affecting the separation efficiency of the particle separator. Jiang [5] analyzed the efficiency of the particle separator under different conditions and revealed the separation mechanism of the particle separator by Fluent. They also researched the effects of particle size, shape factor, resilience characteristics, gravity, particle inlet velocity, inlet mass distribution, and engine operating conditions on the removal efficiency. Barone [6,7,8,9,10] used the Stokes number to express the ratio of particle inertial action to diffusion action, designed and built a device to study the complicated flow of the inertial particle separator, and observed the particle separator using a two-dimensional flow phenomenon. There was an obvious secondary circulation in the scavenge flow path, and the separation efficiency of A4 coarse dust in three designed particle separators was measured by particle image velocimetry (PIV) technology. The separation efficiency of the three particle separators was compared using sand particles with diameters of 10 μm, 35 μm, and 120 μm, and the test results were consistent with the expected range. Large-size particles are greatly affected by inertia, and the separation efficiency of the three particle separators for 120 μm particles is close to 100%. The separation efficiency of the scavenging ratio (SCR) for small particles is important. The Stokes number is a dimensionless constant used to express the ratio of particle inertia to diffusion. A mathematical model to predict separation efficiency is proposed through the Stokes number. In another way, scholars have carried out considerable research on the wall surface of the particle separator. Yuan [11] analyzed the particle trajectories and improved the separation efficiency by modifying the wall material. Li [12] used a combination of experimental and theoretical analysis to study the mechanism of particle rebound by processing an image obtained with a high-speed camera, where the sand rebound coefficient of different materials was established. Through many experiments, Wakeman [13] obtained the collision rebound characteristic formula of sand particles on the surface of a 2024 aluminum alloy. The 2024 aluminum alloy is widely used in the aviation field, and the application of a collision rebound characteristic formula is widely used. Ling [14] arranges a flexible cord/rubber composite material in the central body of a particle separator to change the geometric shape of the device. Snyder [15] has also established a new particle separator configuration and proposed a combined particle separator scheme that divides the airflow into two parts, connects them in parallel, and finally separates them through two upper and lower parallel particle separators.

At present, the particle separator is only useful for particles larger than 10 μm in size, and the separation efficiency of particles smaller than 10 μm in size is still determined by aerodynamic performance, so overall separation efficiency cannot be completely improved. This paper performs numerical calculations on the classic configuration of the integrated particle separator from the scientific and technological reports found in the literature [16,17,18,19,20], and the calculation results are also compared and analyzed with the data from the paper. Through reverse analysis of the particle trajectory, the material of the front wall of the shroud is modified to improve performance. A tandem particle separator structure was designed to improve the overall separation efficiency of the particle separator. The second part of this paper introduces the research object, the third part introduces the computational model and the computational method, the fourth part provides an experimental comparison and validation of the algorithm, and the fifth part proposes two particle separator optimization methods.

## 2. Research Objectives

The world’s first set of IPS was applied to the T700 engine of the “Black Hawk” helicopter, and then the T700 engine equipped with IPS was used in the “Apache” helicopter. The main components of IPS include the center body, outer cover, splitter, scavenging volute, and collection tube, and the structure is rather unique. Under the action of inertia and air, sand particles bounce back to the scavenge flow path through the center body and shroud. Then, the clean air enters the compressor. Figure 1a shows a CH-47 helicopter with the IPS system installed on the engine inlets. Figure 1b illustrates the T700 particle separator. The pre-rotating blades rotate the airflow-containing particles. The particles hit the shroud at a certain angle through the pre-rotating blades, and the particle separator uses inertia to separate the particles.

The particle separator mounted on the engine is an axisymmetric bifurcated tube structure, and a typical T700 engine inlet IPS cross-section schematic is shown in Figure 1c. The fluid area includes an air inlet channel, a scavenge flow path, and a core flow path. The solid structure consists of a shroud, hub, and splitter. There is a bulge in the central body, and its profile is sharply curved along the flow direction. The airflow containing sand undergoes a large angle of deflection under the guidance of the central body bulge, resulting in a large centrifugal force for the airflow containing sand. Due to the large inertia of the sand particles, they are forced to move closer to the outside of the channel and enter the air sweeping channel, while most of the airflow enters the compressor through the main flow channel.

Considering that the particle separator channel is three-dimensional, its circumferential shape is the same, so the simulation calculation simplifies the 3D model into a 2D model for the calculation to conserve calculation resources. The main focus of the calculation results is on the separation efficiency of sand particles and the total pressure recovery coefficient. The research object is an inertial particle separator similar to the one used in the T700 engine. The basic structural sketch of the model is shown in Figure 1d. The basic data and design solutions of the model were obtained from the scientific and technical reports in the references [17].

## 3. The Basic Model and Calculation Method

The simulations in this paper were implemented using the commercial software Fluent, where the gas phase adopts the two-dimensional realizable k-*ε* model and is solved by using a pressure–velocity coupled algorithm. The particle phase is calculated using the discrete phase model (DPM) that comes with Fluent software. The DPM model requires the volume fraction of the particle phase to be less than 10%, and according to the literature [16,17], the inlet particle concentration of helicopter IPS is $5.3\times 10^{-5}$ kg/m^3^ ~ $5.3\times 10^{-4}$ kg/m^3^ and the density of quartz sand is 2650 kg/m^3^. The volume fraction occupied by the particles is easily obtained and is much less than 10%. Therefore, only one-way coupling is considered in the calculation. The fluid domain is calculated first, the particle phase is added after the convergence of the fluid domain, and the trajectory of the solid phase is solved according to the force balance of the particles.

### 3.1. Governing Equations

Fluid flow in a particle separator needs to satisfy the continuity and the momentum conservation equation. The continuity equation is the expression of the law of mass conservation in fluid mechanics, and the momentum conservation equation is the equation of motion describing the conservation of momentum of a viscous fluid, also known as the Navier–Stokes (N-S) equation, both of which are important in fluid mechanics. Fluent uses the finite volume method to discretely solve the N-S equation. In the Cartesian coordinate system, the continuity equation is shown in Equation (1), where $\frac{\partial \rho }{\partial t}$ represents the increase in mass at a point in space and $\rho \vec{V}$ represents the mass flowing out of that point.
(1)$$\frac{\partial \rho }{\partial t}+\nabla \cdot (\rho \vec{V})=0$$

The conservation of momentum equation is shown in Equation (2), where $\frac{D\vec{V}}{Dt}$ is the change in momentum of the fluid with time for the inertial force term, ${\vec{f}}_{b}$ is the volume force term, $-\frac{1}{\rho }\nabla p$ is the differential pressure force term, and $\frac{\mu }{\rho }\nabla^{2}\vec{V}$ is the viscous force term.
(2)$$\frac{D\vec{V}}{Dt}={\vec{f}}_{b}-\frac{1}{\rho }\nabla p+\frac{\mu }{\rho }\nabla^{2}\vec{V}$$

When the fluid is in a turbulent state, the system also has to obey the turbulence model. The simulation in this paper assumes a constant turbulent isothermal flow. The Reynolds number at the inlet of particle separator is 975,240, so the turbulence model uses the realizable *k-ε* model, which is the most well-known and widely used two-equation vortex viscosity model, which has better results for free-shear-layer flows at relatively smaller pressure gradients and for wall-bounded flows at small average pressure gradients, in better agreement with experimental results. Solving the fluid domain using the pressure–velocity coupled SIMPLEC (Semi-Implicit Method for Pressure-Linked Equations Consistent) second-order equation [21]. The *k-ε* model equation is as follows:

*k* Equation:(3)$$\frac{\partial }{\partial t}(\rho k)+\frac{\partial }{\partial x_{j}}\left(\rho ku_{i}\right)=\frac{\partial }{\partial x_{j}}\left[\left(\mu +\frac{\mu_{t}}{\sigma_{k}}\right)\frac{\partial k}{\partial x_{j}}\right]+G_{k}+G_{b}-\rho \varepsilon -Y_{M}+S_{k}$$

*ε* Equation:(4)$$\frac{\partial }{\partial t}(\rho \varepsilon )+\frac{\partial }{\partial x_{i}}\left(\rho au_{i}\right)=\frac{\partial }{\partial x_{j}}\left[\left(\mu +\frac{\mu_{t}}{\sigma_{\varepsilon }}\right)\frac{\partial \varepsilon }{\partial x_{j}}\right]+C_{1\varepsilon }\frac{\varepsilon }{k}\left(G_{k}+C_{3\tau }G_{b}\right)-C_{2\varepsilon }\rho \frac{\varepsilon^{2}}{k}+S_{\varepsilon }$$
where *G_k_* denotes the generation of turbulent kinetic energy by the mean velocity gradient. *G_b_* is the turbulent kinetic energy generated by buoyancy. *Y_M_* denotes the contribution of fluctuating expansion to the total dissipation rate in a compressible turbulent flow. $C_{3\tau }=\tanh\left|u/v\right|$. The turbulence velocity is determined by the equation $\mu_{t}=\rho C_{\mu }k^{2}/\varepsilon$, where $C_{\mu }=0.09$. $C_{1\varepsilon }=1.44$ and $C_{2\varepsilon }=1.92$ are constants. $\sigma_{k}=1.0$ and $\sigma_{\varepsilon }=1.3$ are Prandtl numbers. The turbulent kinetic energy k = 8.3 at particle separator operating conditions, and the turbulent dissipation rate *ε* = 0.007.

### 3.2. Aerodynamic Model

In the interaction between particles and viscous fluids, particle motion is usually subject to gravity, buoyancy, traction, Magnus force, etc. Among them, the buoyancy force, pressure gradient force, and Basset force are much smaller than the inertia force and neglected, the false mass force is much smaller than the density of air, and the relative acceleration of particles is not large, so it is also neglected, and the Saffman force is only considered in the boundary layer due to the small velocity gradient in the mainstream region. The Saffman force is only considered in the boundary layer due to the small velocity gradient in the mainstream region. Although the forces acting on the particles are quite complex, the drag force is still the most important force, which plays an important role in the flow and heat transfer. In the calculation process, the force of sand particles on the flow field and collisions between particles are neglected because the volume of solid particles in the air is small and the trajectory of sand particles depends on the gravity of sand particles, traction forces, and collisions with walls. The DPM model is calculated based on the Eulerian–Lagrange trajectory method, where the fluid is described in the Eulerian equations as a continuous medium and the particles as discrete phases using the Lagrange trajectory model, and the trajectory tracking is accomplished by differentiating the equations of motion for the integrated particles. Traction force per unit mass of particles can be calculated using Equation (5).
(5)Fd=18μρpdp2⋅CdRe24
where μ is the fluid viscosity, $\rho_{p}$ is the density of sand, $d_{p}$ is the diameter of the particles, and Re=ρdpup−uμ is the relative Reynolds number, where *u* is the air velocity and *ρ* is the airflow density. The Reynolds number range of AC-Coarse is 1.26–252, and the Reynolds number range of C-Spec is 1.26–1260. In this paper, the spherical drag law is used and the drag coefficient for smooth particles can be derived from Cd=a1+a2/Re+a3/Re2, where $a_{1}$, $a_{2}$, and $a_{3}$ are constants with values determined by the Reynolds number range. Morsi [22] measured the values $a_{1}$,$a_{2}$, and $a_{3}$ by analyzing standard test data approximations for spherical particles. The corresponding drag coefficients for different particle Reynolds numbers are shown below.
a1,a2,a3=0,24,00<Re<0.13.690,22.73,0.09030.1<Re<11.222,29.1667,−3.88891<Re<100.6167,46.50,−116.6710<Re<1000.3644,98.33,−2778100<Re<10000.357,148.62,−475001000<Re<50000.46,−490.546,5787005000<Re<100000.5191,−1662.5,5416700Re≥10000

### 3.3. Particle Wall Rebound Model

At present, most of the rebound characteristics of the wall are empirical formulas based on statistics and fitting of a large number of experimental results. A large number of experimental results by Wakeman and Tabakoff [13] show that when quartz particles rebound on an annealed 2024 aluminum alloy plate, the statistical performance of the rebound characteristics can be described by Equation (6).
(6)$$V_{n2}/V_{n1}=0.993-0.0306\beta_{1}+0.00047\beta_{1}^{2}-2.58e^{-6}\beta_{1}^{3}\;V_{t2}/V_{t1}=0.988-0.0289\beta_{1}+0.000642\beta_{1}^{2}-3.53e^{-6}\beta_{1}^{3}$$

The 45-steel is a high-quality carbon structural steel. Tabakoff [23] used a similar method to obtain a mathematical model for the rebound of particles on a 45-steel, which can be described by Equation (7).
(7)$$V_{n2}/V_{n1}=0.9031-0.02136\beta_{1}+0.0004023\beta_{1}^{2}-9.315e^{-7}\beta_{1}^{3}\;V_{t2}/V_{t1}=0.9803-0.02846\beta_{1}+0.0004738\beta_{1}^{2}-2.454e^{-6}\beta_{1}^{3}$$
where *β*_1_ and *β*_2_ are the incident angles; the unit is the angular measure. As shown in Figure 2, *V_n_*_1_ and *V_n_*_2_ are the normal components of the incident and bounce velocities, respectively. *V_t_*_1_ and *V_t_*_1_ are the tangential components of the incident and bounce velocities, respectively.

### 3.4. Performance Parameters

To have a uniform distribution of particles along the radial direction at the inlet, the particle delivery method was set as a surface jet source, and the particles were delivered from the inlet. When calculating the trajectory of a single particle size, a single distribution was chosen. When calculating the trajectories for AC-Coarse and C-Spec, the sand particle distribution is the Rosin–Rammler distribution, which can be easily defined by fitting the particle size distribution data to the Rosin–Rammler equation. In this approach, the complete particle size range is divided into a set of discrete particle size ranges. There is an exponential relationship between the particle diameter *ds* and the mass fraction *Y_d_* of particles larger than the specified diameter, Equation (8).
(8)$$Y_{d}=\exp^{-{\left(d_{p}/d_{s}\right)}^{n}}$$
where $d_{p}$ represents the particle size of the particle; $Y_{d}$ represents the mass fraction of particles with a particle size less than $d_{p}$; $d_{s}$ represents the average particle size of a particle; *n* represents the distribution index of the particle.

According to military sand control requirements, the sand particles that must be considered include “MIL-E-5007C,” known as C-Spec, and “Coarse Arizona Road Dust,” known as AC-Coarse. The particle size distribution data of the two types of mixed sand are shown in Table 1.

Separation efficiency is an important indicator of IPS performance. It is defined as the ratio of the mass of sand in the scavenge region to the mass of sand particles contained in the airflow in the inlet region and is an important indicator for judging the purified air of the particle separator. It can be described by Equation (9).
(9)$$\eta_{W}=\frac{M_{0}-M_{1}}{M_{0}}\times 100\%$$
where *M*_0_ is the mass of sand and dust entering the inlet flow path of the particle separator per unit time, and *M*_1_ is the mass of sand and dust entering the compressor from the inlet flow path per unit time.

The SCR is an important factor that determines the separation efficiency, and it is also a bridge that converts the mass separation efficiency to the concentration separation efficiency. It can be described by Equation (10).
(10)$$SCR=\frac{w_{2}}{w_{1}}$$

In Equation (10), *w*_2_ is the scavenge mass flow, and *w*_1_ is the core mass flow. The separation efficiency increases as the SCR increases, but when the SCR increases to about 22%, the increase in separation efficiency is not obvious, so the SCR generally ranges from 16% to 22%.

The total pressure recovery coefficient is an important indicator used to evaluate the aerodynamic loss of airflow through the channel. The core airflow of the particle separator goes directly to the compressor, and the core mass flow is directly related to the working state of the engine. Therefore, the total pressure recovery coefficient of the core region’s outlet is an important indicator for evaluating the aerodynamic performance of the particle separator. The total pressure recovery coefficient can be described by Equation (11).
(11)$$\Delta P=\frac{P_{1}}{P_{0}}\times 100\%$$
where *P*_0_ is the total pressure of the inlet region, *P*_1_ is the total pressure of the core region, and the ratio of the total pressure loss to the total pressure of the inlet region is the total pressure loss coefficient.

## 4. Verification of Calculation Examples

### 4.1. Mesh Generation

We used Pointwise meshing software to mesh the established reference physical model. The entire computing area uses a structured grid for meshing. Since the simulation uses the *k-ε* model for calculation, the grid is encrypted near the wall surface to ensure that the boundary y+ is above 30.

In this work, the mesh-independence of the particle separator was verified, as shown in Figure 3. The left side of Figure 3 shows the particle separator mesh scales of 30,000, 70,000, 100,000, and 150,000, and the right side of Figure 3 shows the calculation results of different scales of the mesh using Fluent’s DPM model for AC-Coarse and C-Spec. From the calculation results, it can be seen that the error in the separation efficiency of the two particles after increasing the mesh scale separation does not exceed 2.5%. In this paper, the particle separator simulation mesh scale is determined to be 30,000.

### 4.2. Boundary Conditions

To verify the correctness of the simulation methods used in this article, the simulation was carried out based on the actual experimental conditions with a mass flow rate of 2.28 kg/s and an SCR of 18%. Suppose the flow field is a stable isothermal symmetric flow field, and set the calculation residual to 1 × 10^−5^ and the calculation step size to 1500. The medium is a gas–solid two-phase flow, the air density is 1.225 kg/m^3^, and the viscosity is 1.7894 × 10^−5^ Pa·s. The sand is selected from quartz sand particles, and the density is 2650 kg/m^3^. They are spherical particles that are not deformed or do not collide with each other. Two common simulation boundary conditions are used to analyze the flow field of the particle separator, and the simulation and experimental results are compared in these two cases.

The boundary conditions of the fluid domain are shown in Table 2.

Boundary condition 1

Simulation 1 inlet boundary using total pressure inlet, the scavenge flow path and the core flow path are mass flow outlets, and the mass flow ratio between the scavenge flow path and the core flow path is 0.18.

Boundary condition 2

Simulation 2 inlet boundary using speed inlet, the scavenge flow path is set to a pressure outlet size of P_0_, the core flow path is also a pressure outlet, and the pressure magnitude is set to 0.82 × P_0_.

The discrete phase boundary conditions are as follows:Boundary condition 1:

The sand velocity is set to 80% of the air velocity. The inlet section velocity of boundary condition 1 is 87.14 m/s after surface average integration. The sand inlet velocity is set at 69.6 m/s. The grit inlet angle is set to 36 degrees because the T700 particle separator has pre-rotating blades, which use inertia to separate particles after they hit the shroud. V_x_ = 55.68 m/s and V_y_ = 41.94 m/s.

Boundary condition 2

The velocity inlet is set to 84.5 m/s, and the sand velocity is set to 80% of the air velocity at an inlet angle of 36 degrees. The sand velocity is set to 67.6 m/s. V_x_ = 54.04 m/s, and V_y_ = 40.56 m/s.

### 4.3. Flow Field Analysis

Figure 4 shows the particle trajectory and residence time for the AC-Coarse and C-Spec separation processes using the two boundary conditions. There is not a significant difference in the separation effects of the two boundary conditions through the trajectory diagrams, but it can be seen from the running time of the two particles that AC-Coarse stays in the IPS for a longer time, with the longest time being 0.0041 s, while C-Spec stays in the IPS for a shorter time, with the longest time being 0.00183 s. It can be seen that the trajectory of AC-Coarse looks more chaotic, while the trajectory of C-Spec looks more regular. The trajectory of AC-Coarse has curves while the trajectory of C-Spec is straight. It can be seen that when the AC-Coarse is separated, there is a vortex at the upper end of the scavenge flow path, resulting in secondary separation, and since the average quality of AC-Coarse is lower than that of C-Spec, it is highly affected by the force of the airflow. At the air vortex, some AC-Coarse is re-exported to the separate flow path, and part of the AC-Coarse is sucked into the core flow path. Therefore, the cleaning time is longer. Compared with AC-Coarse, C-Spec is more strongly reflected by the wall and less affected by airflow. It repeatedly collides with the wall into the scavenge flow path, and the rebound force of the wall is more obvious. Therefore, the cleaning time is shorter, and it has a better cleaning effect on C-Spec.

Figure 5 and Figure 6 show the static pressure diagram and wall static pressure curve diagram under the two boundary conditions, respectively. By comparing the experimental data with the simulation data in the literature, the static pressure distribution is shown in Figure 5. The hydrostatic pressure distribution of Simulation 1 has a high similarity to the hydrostatic pressure distribution in the literature, and the hydrostatic pressure distribution of Simulation 2 has relatively low similarity to the hydrostatic pressure distribution in the literature. Figure 6 shows the wall static pressure curve. It can also be seen that at the center body, the static pressure value of Simulation 1 is around 90,000 Pa, the static pressure value of Simulation 2 is around 85,000 Pa, and at the entrance of the inlet path, the maximum static pressure at the boundary of Simulation 2 has exceeded 110,000 Pa, and the error is relatively large. The static pressure distribution for the shroud and the center body of Simulation 1 has a smaller error compared with the experimental value [17]. Therefore, the calculation results of the shroud and hub for Simulation 1 are closer to the experimental result compared to Simulation 2. It can be judged that boundary condition 1 is more accurate for the simulation of static pressure and more suitable for the simulation research of particle separators.

Figure 7 shows the Mach number cloud map and a velocity vector diagram of the two boundary conditions. From the Mach number cloud map, it can be seen that the maximum Mach number of boundary condition 2 at the center body bulge reached 0.63, and then it can be seen that the maximum Mach number of boundary condition 1 at the bulge of the central body reaches a maximum Mach number of 0.44, which is also the reason for the higher static pressure at the center body of boundary condition 1. It can be seen from the velocity vector diagram that in the upper part of the scavenge flow path, the velocity is close to 0, and there is a vortex that produces a secondary separation effect. Both boundary conditions will produce vortex airflow, and the vortex airflow will cause the particle separation time to be longer. In addition, some of the small particles will be sucked into the core flow path. The gas velocity began to increase at the bulge of the central body, and the highest Mach number of the two boundary conditions reached 0.63 at the cross-section of the core flow path, but the distribution of boundary condition 1 was more uniform. The calculation results for boundary condition 1 are closer to the literature.

Boundary condition 1: The total outlet pressure at the core flow path is 105,550.9 Pa after mass-weighted average integration.

Boundary condition 2: Inlet mass flow calculation result is 2.33 kg/s, scavenge path outlet mass flow is 0.47 kg/s, core path outlet mass flow is 1.85 kg/s, and the SCR was calculated to be 25.4%. The total inlet pressure is 107,170 Pa and the total outlet pressure of the core mass path is 106,340 Pa after mass-weighted average integration.

The calculation results are shown in Table 3. The total pressure recovery coefficient of boundary condition 1 is 99%, and the total pressure recovery coefficient of boundary condition 2 is 99.2%. The calculation results of the two simulations are close to the experimental values. For AC-Coarse with boundary condition 1, the separation efficiency is 82.5% and the separation efficiency of C-Spec is 90.9%, while the separation efficiency of boundary condition 2 AC-Coarse is 82.1% and the separation efficiency of C-Spec is 88.2%. It can be seen that the calculation result of boundary condition 1 is in better agreement with the experimental value than the calculation result of boundary condition 2. Where the SCR value of boundary condition two is 25.4%. Therefore, it is recommended to use the boundary conditions of the total inlet pressure and the flow outlet in the numerical calculation of IPS. Not only can SCR be controlled, but also the total pressure recovery coefficient and the separation efficiency of the two particles are close to the experimental values. Therefore, the calculation method used in Simulation 1 has a high degree of confidence and can be used for flow field simulation.

## 5. Wall Surface Optimization and IPS Modeling Optimization

### 5.1. Particle Trajectory Analysis

Simulations were carried out to calculate the trajectories for a wide range of particle sizes. Figure 8 shows the trajectories and residence times of 1 µm, 5 µm, 10 µm, 50 µm, 100 µm, and 300 µm particles in the flow domain. The smaller particles have fewer collisions with the wall, and the larger particles have multiple collisions with the wall, as would be expected. For the smaller particles, the dragging force of the airflow overcomes the inertia of the particles, thus avoiding collisions with the walls. As the particle size increases, the drag force on the particle’s inertia decreases, and the particle crosses the airflow channel to collide with the wall, the number of bounces being related to the bounce coefficient of the wall material and also to the size of the particle. It is worth noting that the separation efficiency of particles with a diameter of 10 µm is 100%, with a portion of the process entering the scavenging flow path colliding with the wall and another portion recirculating near the scavenging flow. As seen in Figure 7, there is a low-pressure zone in the scavenging flow path, where a portion of the 10 µm particles are reflected by inertia into the scavenging flow channel. The airflow does work on the particles, the initial velocity of the particles decreases, and they follow the airflow completely under the action of the trailing force, which also corresponds to the velocity vector diagram, and thus the 10 um particles produce a unique vortex phenomenon.

Figure 9 shows the distribution of different diameters of AC-Coarse particles and C-Spec particles in the fluid channel, which corresponds to the trajectories of particles of different sizes above. AC-Coarse has more small particles below 10 µm than C-Spec, so more particle recycling can be seen in the scavenging flow path. Only part of the AC-Coarse small particles enters the core stream path, and most of the particles entering the core stream path are reflected by the first reflection at the front end of the shield and the second reflection at the beam splitter. The particle size of C-Spec is mainly distributed in the hundreds of microns. These particles are less affected by the drag force of the airflow and have higher kinetic energy. They enter the core stream through the reflection of the shroud. Through the analysis of particle trajectories, two design schemes to improve the separation efficiency of IPS are proposed and simulated.

### 5.2. Wall Surface Optimization Simulation

By reverse tracking the particles entering the core flow path, it can be seen that some particles are reflected by the wall at the front end of the shroud and then directly enter the core flow path. This is caused by the rebound characteristics of the 2024 aluminum alloy. The 2024 aluminum alloy at the front end of the shroud is then adjusted to 45-steel. The position of the replacement material is shown in Figure 10. The particle bounce equation for the 45-steel material in the simulation is Equation (8).

Figure 11 shows the particle trajectory and residence time of AC-Coarse and C-Spec using a particle separator with replacement front wall material of 45-steel. The rebound characteristics of 45-steel and 2024 aluminum alloy are inconsistent. The 45-steel material reflects particles that would otherwise be reflected into the core flow path back to the center body, and the particles are reflected several times into the scavenge flow path. It can be seen from the trajectory diagram that the flow of particles into the core flow path has been reduced. After replacing the material with 45-steel, the separation efficiency of AC-Coarse was 93.3%, an increase of 12.4% under the same boundary conditions as with the original shroud material, while the separation efficiency of C-Spec was 97.6%, an improvement of 6.1%. Changing the wall material not only improves the separation efficiency of AC-Coarse and C-Spec but also does not affect the total pressure loss.

Simulation calculations were performed on spherical particles of 50 µm, 100 µm, and 300 µm. The calculated results are shown in Figure 12. It can be seen that compared to 2024 aluminum alloy, more particles of 50 µm will be affected by the upper wall materials and enter the core flow path. Only a small fraction of the 100 µm particles enter the core flow path, and the separation effect is better than that of the 2024 aluminum alloy. The 300 µm particles rebounded to the center body, rebounded back to the shroud, and finally entered the scavenge flow path. Only a portion of the particles entered the core flow path. It can be seen that replacing the wall material is effective, allowing more particles above 50 µm to be reflected into the scavenge flow path while also increasing the number of small particles below 50 µm entering the core flow path, which is beneficial as a whole. Large particles will cause more damage to the compressor, and the overall separation efficiency will also increase after replacing the wall material. Therefore, replacing part of the wall material is beneficial to improving the performance of the helicopter.

### 5.3. IPS Modeling Optimization

A new idea with potential application to particle separators is proposed. Figure 13 shows a tandem IPS model structure for the secondary separation of sand particles. A new particle separation channel was added to the tail of the previous T700 particle separator, which was not designed in detail in this paper and can be optimized for subsequent design. Secondary separation is used to improve the separation efficiency of IPS and modify the core flow path of the original particle separator into another new IPS. The second bulge on the central body is used to separate the sand particles entering the core flow path. The simulation adopts the boundary conditions of a total inlet pressure of 108,000 Pa and the flow outlet. The ratio of the flow rate of the first scavenging airflow path to the total flow rate of the second particle separator is 18%, the SCR of the second particle separator is 18%, and the flow rate of the core flow outlet is 1.93 kg/s. This ensures that it is consistent with the core flow outlet of the experiment.

Figure 14 shows the trajectories of AC-Coarse and C-Spec in the tandem IPS. It can be seen that when AC-Coarse and C-Spec pass through the second bulge on the central body, they enter the second scavenge flow path by multiple reflections. The separation efficiency of the secondary protected IPS for AC-Coarse was 91.7%, which is an increase of 10.8% under similar conditions without the secondary protection unit. The separation efficiency of C-Spec is 97.7%, which is an increase of 6.2% under similar conditions without a secondary protection unit, but the total pressure loss is 3.3%, which slightly exceeds the design requirement of less than 3%. In the next stage, the configuration can be optimized to reduce the total pressure loss under the premise of ensuring separation efficiency.

Figure 15 shows the 50 µm, 100 µm, and 300 µm particle trajectories. It can be seen that most of the 50 µm particles have been separated when they enter the first layer of the particle separator, and the remaining particles are separated into the second layer scavenge flow path. The 100 µm particles rebound from the front of the shroud into the second layer of the particle separator and then rebound at the center body bulge of the second layer. Some particles still enter the core flow path, and most of the particles are still separated by the second layer of the particle separator. The 300 µm particle undergoes two particle separations, and a small portion of the separated particles enters the core flow path. Figure 15 shows that only some of the particles in the range of 50 µm to 300 µm will enter the core flow path. The particles entering the core flow path are mainly concentrated around 100 µm.

Secondary separation of particles is effective. The secondary protection configuration of IPS can separate the sand and dust in the airflow, but the shortcomings are also obvious. It will increase the size and weight of the particle separator, which will make the layout of the IPS on the helicopter more difficult. The total pressure loss is also large, but there is still room for optimization. Although the secondary particle separator has more drawbacks, it is still a new idea to remove particles from the engine, and the next stage can be to optimize its design in detail so that it can be applied to the engine.

## 6. Conclusions

(1) The simulated boundary conditions of the particle separator were studied, and it was found that the simulated results of the boundary conditions of the total inlet pressure and the mass flow outlet were more accurate and closer to the experimental values.

(2) By changing the 2024 aluminum alloy at the front end of the shroud inlet to 45-steel, the rebound trajectories of some particles were changed so that these particles were reflected from the bulge of the central body into the scavenge flow path. Among them, the separation efficiency of AC-Coarse was 93.3%, an increase of 12.4%, while the separation efficiency of C-Spec was 97.6%, an increase of 6.1%. The main particle size range of particles entering the core flow path was around 50 µm.

(3) A tandem IPS structure was established to improve the separation efficiency of IPS through a multi-stage separation configuration so that the particles can effectively improve separation efficiency of the particles after multiple rebounds. The separation efficiency of AC-Coarse was 91.7%, an increase of 10.8%, while the separation efficiency of C-Spec was 97.7%, an increase of 6.2%. However, the total pressure loss was 3.3%, and the main particle size range of particles entering the core flow path was around 100 µm. The next step in the research will be an optimization to decrease the separation efficiency to less than 3% to meet the requirements of IPS total pressure loss.

## Figures and Tables

**Figure 1 entropy-25-00147-f001:**
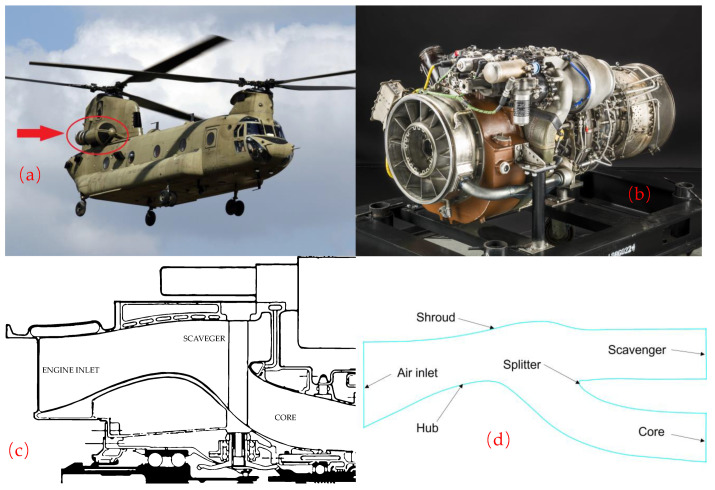
CH-47 helicopter with an T700 system installed. (**a**) was modified from Flickr (https://www.flickr.com/ (Accessed: 1 December 2022)), licensed under a Creative Common At-tribution 2.0 Generic License (https://creativecommons.org/licenses/by/2.0/ (Accessed: 1 December 2022)). (**b**) has been marked as dedicated to the public domain, from (https://wordpress.org/openverse/ (Accessed: 1 December 2022)). Licensed under CC0 1.0 universal public domain (https://creativecommons.org/publicdomain/zero/1.0/ (Accessed: 1 December 2022)). (**c**) Schematic of a typical T700 engine inlet IPS. (**d**) Sketch of T700 particle separator model.

**Figure 2 entropy-25-00147-f002:**
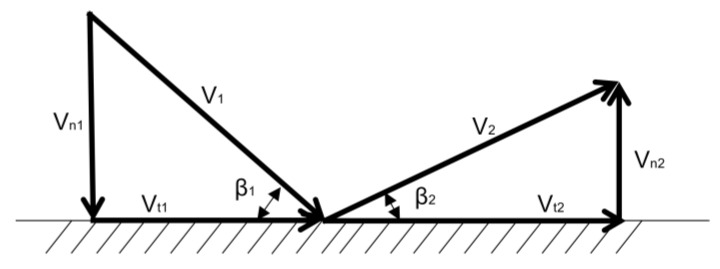
Schematic diagram of sand particles colliding and rebounding on the wall.

**Figure 3 entropy-25-00147-f003:**
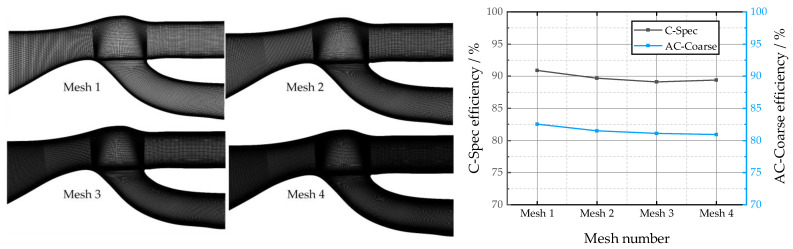
Mesh-independence verification (the scale of mesh 1 is 30,000, the scale of mesh 2 is 70,000, the scale of mesh 3 is 100,000, and the scale of mesh 4 is 150,000).

**Figure 4 entropy-25-00147-f004:**
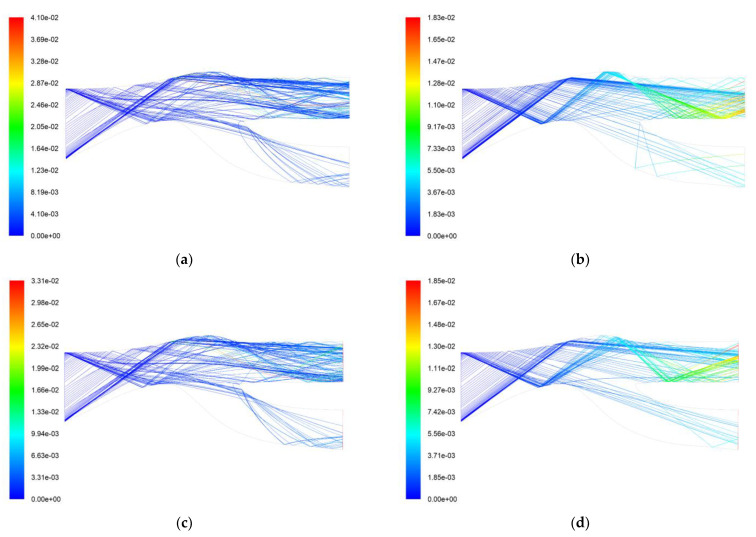
Two kinds of sand separation trajectory and residence time. (**a**) Simulation 1 AC-Coarse trajectory and residence time (s); (**b**) Simulation 1 C-Spec trajectory and residence time (s); (**c**) Simulation 2 AC-Coarse trajectory and residence time (s); (**d**) Simulation 2 C-Spec trajectory and residence time (s).

**Figure 5 entropy-25-00147-f005:**
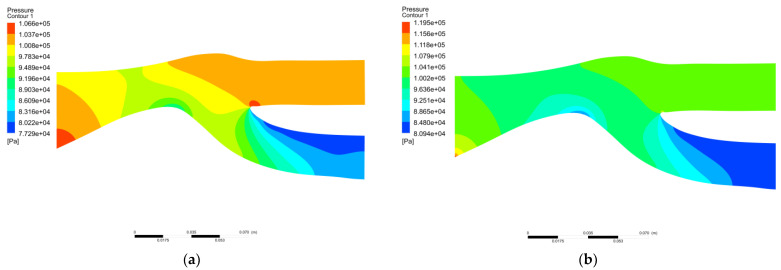
Static pressure distribution diagram. (**a**) Simulation 1 static pressure figure; (**b**) Simulation 2 static pressure figure.

**Figure 6 entropy-25-00147-f006:**
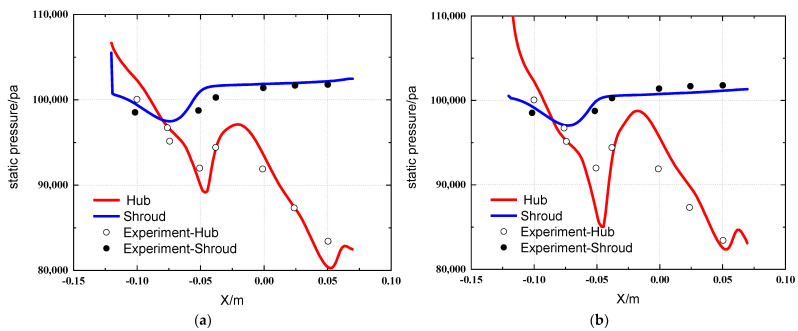
Static pressure curve of the wall. (**a**) Simulation 1 wall static pressure curve; (**b**) Simulation 2 wall static pressure curve.

**Figure 7 entropy-25-00147-f007:**
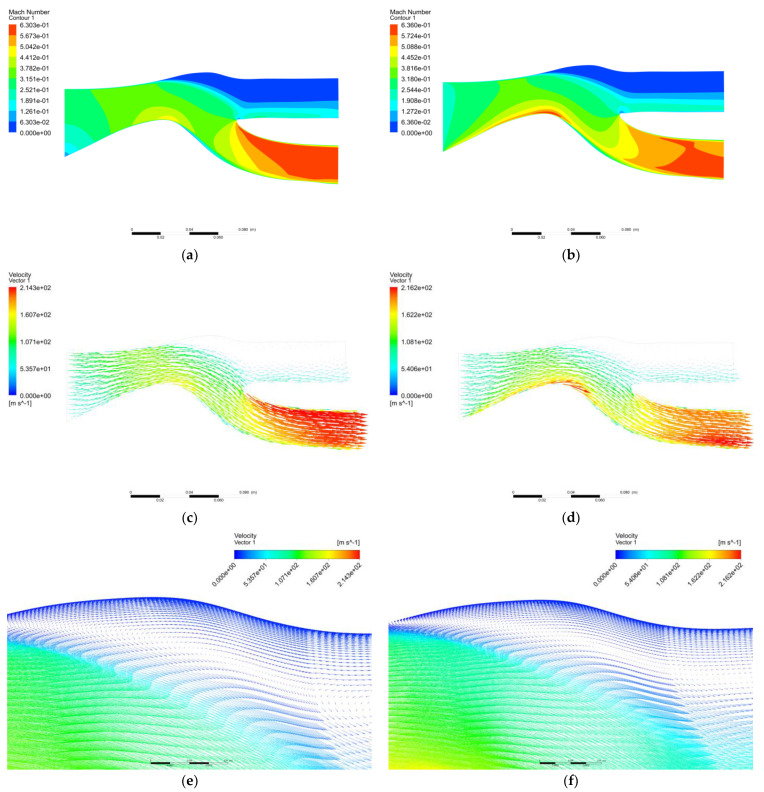
Mach number cloud graph and trace graph. (**a**) Simulation 1 Mach number cloud map; (**b**) Simulation 2 Mach number cloud map; (**c**) Simulation 1 velocity vector diagram; (**d**) Simulation 2 velocity vector diagram; (**e**) Simulation 1 velocity vector diagram; (**f**) Simulation 2 velocity vector diagram.

**Figure 8 entropy-25-00147-f008:**
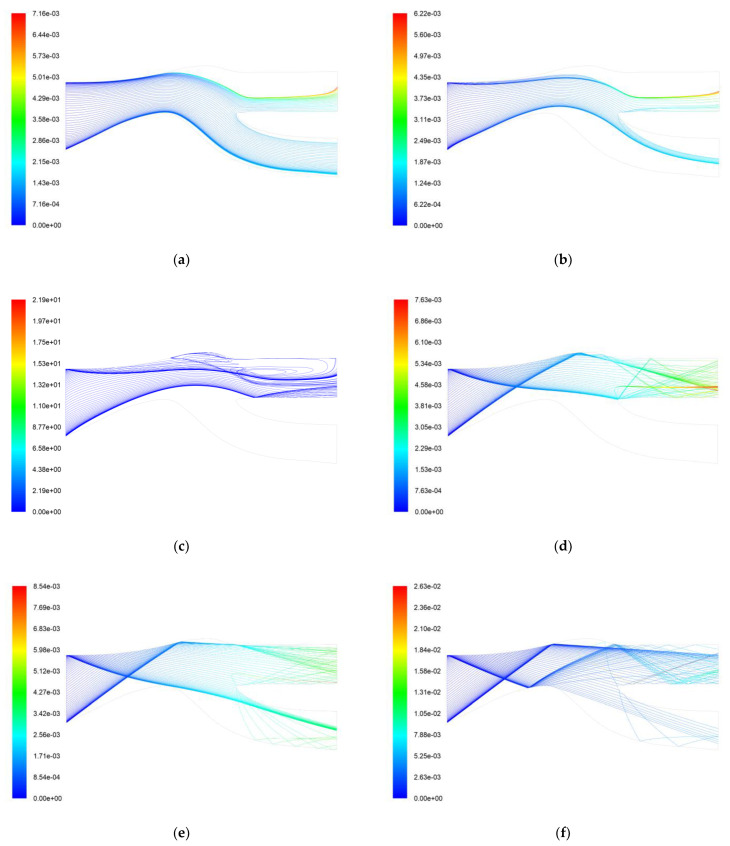
Trajectories and residence time of particles with different diameters. (**a**) 1 µm particle trajectory and residence time (s); (**b**) 5 µm particle trajectory and residence time (s); (**c**) 10 µm particle trajectory and residence time (s); (**d**) 50 µm particle trajectory and residence time (s); (**e**) 100 µm particle trajectory and residence time (s); (**f**) 300 µm particle trajectory and residence time (s).

**Figure 9 entropy-25-00147-f009:**
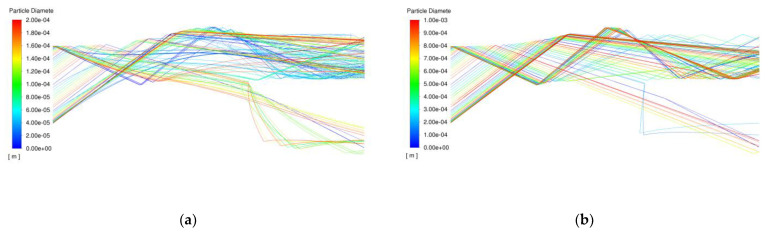
Particle trajectories of AC-Coarse and C-Spec. (**a**) AC-Coarse particle trajectory; (**b**) C-Spec particle trajectory.

**Figure 10 entropy-25-00147-f010:**
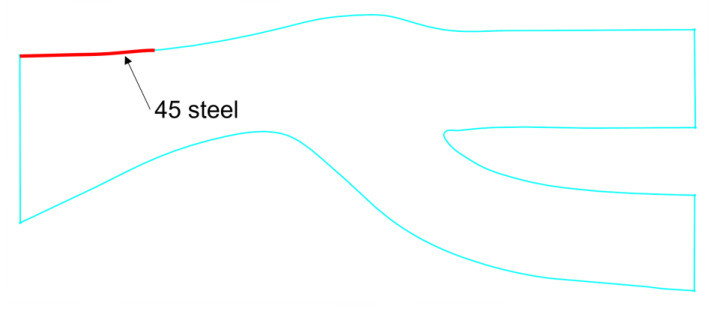
The front end of the shroud is replaced with 45-steel.

**Figure 11 entropy-25-00147-f011:**
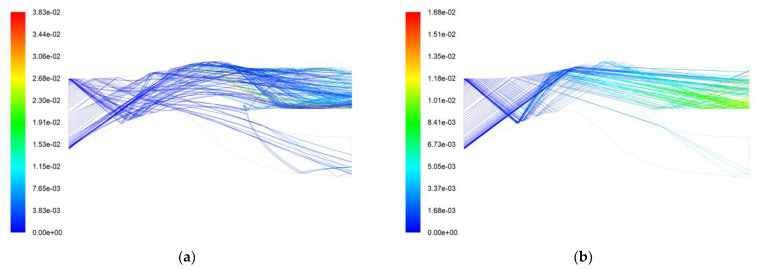
Trajectories and residence time (s) of the two particles. (**a**) AC-Coarse particle trajectory and residence time (s); (**b**) C-Spec particle trajectory and residence time (s).

**Figure 12 entropy-25-00147-f012:**
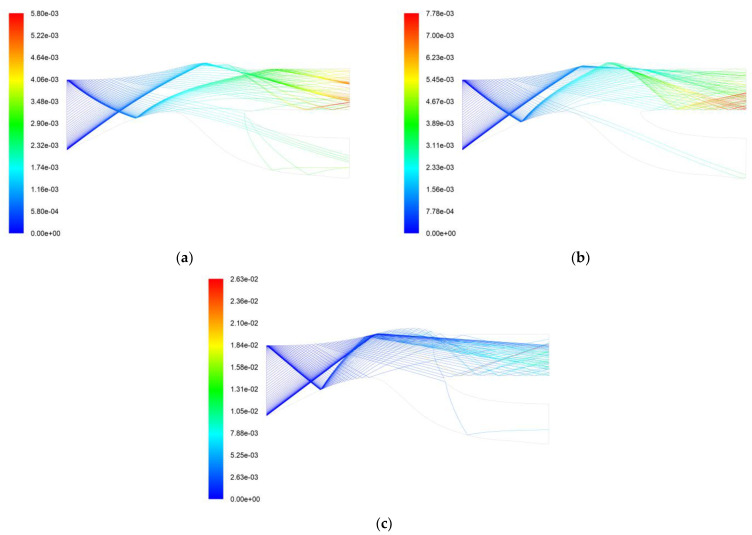
Replacement of 45-steel particle trajectory and residence time. (**a**) 50 µm particle trajectory and residence time (s); (**b**) 100 µm particle trajectory and residence time (s); (**c**) 300 µm particle trajectory and residence time (s).

**Figure 13 entropy-25-00147-f013:**
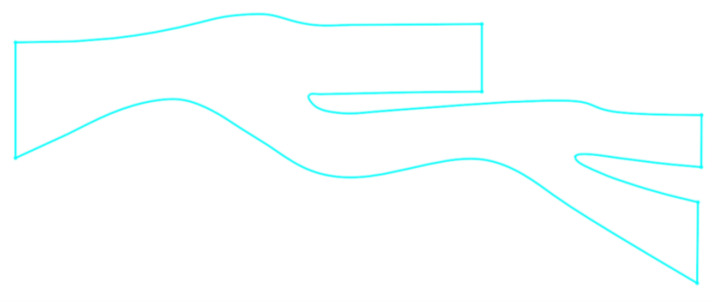
Schematic diagram of the structure of the two-stage tandem IPS.

**Figure 14 entropy-25-00147-f014:**
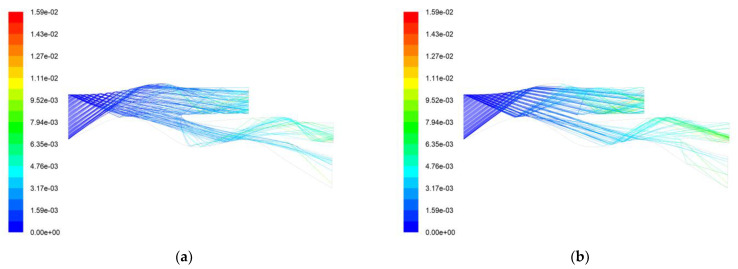
Trajectories and residence time of the two particles. (**a**) AC-Coarse particle trajectory and residence time (s); (**b**) C-Spec particle trajectory and residence time (s).

**Figure 15 entropy-25-00147-f015:**
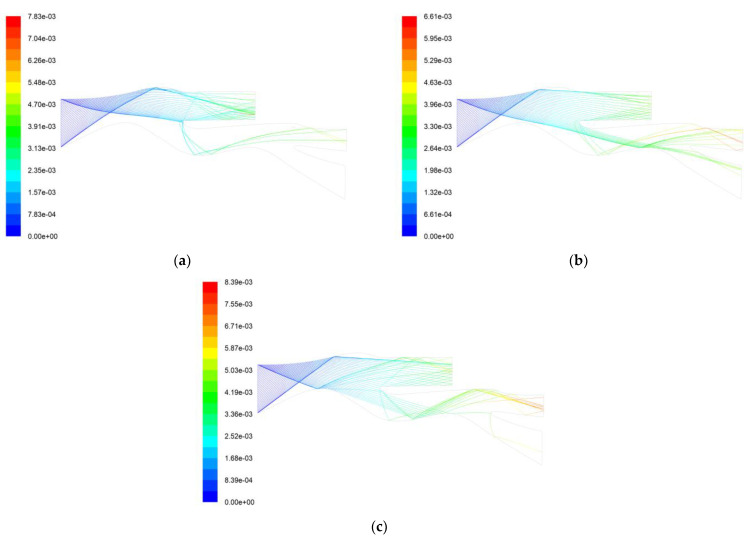
Particle trajectory and residence time of a two-stage tandem IPS. (**a**) 50 µm particle trajectory and residence time (s); (**b**) 100 µm particle trajectory and residence time (s); (**c**) 300 µm particle trajectory and residence time (s).

**Table 1 entropy-25-00147-t001:** Basic parameter table of typical standard sand and dust test sand.

Type	Minimum Particle Size/µm	Maximum Particle Size/µm	Average Particle Size/µm	Spread Parameter
AC-Coarse	1	200	35	1.1
C-Spec	1	1000	310	1.9

**Table 2 entropy-25-00147-t002:** Boundary conditions of the fluid domain.

Type	Inlet	Scavenge	Core
Simulation 1	Pressure Inlet (106,662 Pa)	Mass Flow Outlet (0.35 kg/s)	Mass Flow Outlet (1.93 kg/s)
Simulation 2	Velocity Inlet (84.5 m/s)	Pressure Outlet (101,325 Pa)	Pressure Outlet (83,086.5 Pa)

**Table 3 entropy-25-00147-t003:** Calculation results.

Type	SCR (%)	Total Pressure Recovery Coefficient (%)	Sand Separation Efficiency	
			C (%)	AC (%)
Simulation 1	18.0	99	90.9	82.5
Simulation 2	25.4	99.2	88.2	82.1
Experiment	18.0	99.1	91.5	80.9

## Data Availability

Not applicable.

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
