# Peer review of "Sand Discharge Simulation and Flow Path Optimization of a Particle Separator"

_entropy, 2023, doi:10.3390/e25010147_

Round 1

Reviewer 1 Report

Review of manuscript titled "Sand Discharge Simulation and Flow Path Optimization of a Particle Separator" by Zhou Du, Yulin Ma, Quanyong Xu and Feng Wu for publication at Entropy

This paper presents a CFD study on the separation efficiency of a particle separator in helicopters. The study was done using CFD-DPM methodology, using ANSYS FLUENT. While the topic is certainly interesting, I advice the authors to spend more time on the basics of the CFD analysis, as many of the details presented in the manuscript are either erroneous or missing.

Here are my criticisms regarding the methodology and analytical parts of the manuscript:

1. The Navier-Stokes equations (equation 1) are not Navier-Stokes equations. This should present continuity and momentum conservation equations.

2. Basic Stokes equation should not be quoted to another literature (equation 4)

3. What is the range of particle Reynolds number considered here to warant the use of the Stokes equation?

4. What is the range of Reynolds number of the flow to warant the use of k-epsilon (which is, by the way, not k-e model [line 153])? I realised this should be quite high, given the inlet velocity, but it is good practice to mention this important number.

5. Grid independence study should be presented in the full (not just mentioned).

6. Where are the boundary conditions for the k and epsilon values? 

7. The selection of the boundary conditions are rather odd. I understand that BC 1 uses pressure inlet and mass flow ratio for the 2 outlets, whereas BC 2 uses velocity inlet and pressure values at the outlet. But, this cannot be used for direct comparison against published data on an actual 3D flow separator. Direct claim that 1 BC is better than the other would require a careful consideration of values that have been used, as well as sensitivity analysis on a range of parameters that have been assumed. Only then, you'd be able to claim that 1 BC represents a better idealisation than the other.

8. For a study representing particle separation, why is there no particle sizing parameter in the list of performance parameters?

9. Is the simulation done under steady state or unsteady state? This is of course very important bit of detail that was missing. Also, how do you qualify convergence? What time steps were used?

The analysis of the particle trajectory needs to be done more systematically and scientifically than paths being "straight" or "curves". Again, if you consider particle sizing as one of the performance parameter, you may be able to present numbers rather than just path curves such as figures 7, 11, 13, etc. 

I don't think we can identify vortices based on the velocity vector diagram in figure 10. 

Reviewer 2 Report

The authors present analytic simulations of sand particles of various sizes in an integer particle separator. They validate their computations with experimental measurements and proceed with suggestions that can enhance the efficiency of a particle separator. The first one is a modification of the material at the front of the shroud (which seems more feasible), and the second one proposes a configuration modification with dual protection. 

In general, this is an interesting work, and I suggest its acceptance after minor revisions, which are listed below:

1. The N-S equation shown in Eq. 1 is incorrect. This is the general balance equation for a variable, \phi. There is no description of the terms involved, e.g. S_\phi which denotes the source term, \Gamma_\phi which denotes the flux term, and U should be denoted with a vector sign.

There should be a more precise explanation of the terms involved and of course change the general variable, \phi, to velocity.

2. The authors apply the k-\epsilon realizable model. I believe simulations using different turbulence schemes could also be helpful for the interested reader. 

3. The selection of the different boundary conditions scenarios (for both the fluid domain and the solid phase) is not clear to me. Do the authors select these boundary conditions based on experimental conditions? What is the rationale for selecting these boundary conditions?

Round 2

Reviewer 1 Report

Review of revised manuscript titled "Sand Discharge Simulation and Flow Path Optimization of a Particle Separator" by Zhou Du, Yulin Ma, Quanyong Xu and Feng Wu for publication at Entropy.

This paper presents a CFD study on the separation efficiency of a particle separator in helicopters. The study was done using CFD-DPM methodology, using ANSYS FLUENT. While the authors have answered all my questions in the previous review, I find some of the revisions/modifications minimal and therefore still require further improvement prior to acceptance for publication.

Here are my comments regarding the CFD part of the work:

1. The fact that the particle Reynolds number ranges from 1-1000 indicates that Stokes drag relation cannot be applied throughout. Another correlation that accounts for the full range of particle diameters need to be used.

2. The boundary conditions for k and epsilon (k=0 and ε=0) at the wall are not valid. For the given y+ value, you need to include the use of wall treatments to account for the viscous sublayers at the boundaries. Please read through the FLUENT user manual for these.

3. The discussion on particle tracks is still very lacking and very basic. Yes, the tracks are coloured by residence time, but then what do we make of this? Can this be presented based on average residence times in the different outlets? What about average particle diameter & mass at each outlet?

4. The particle track of the 10 micron particles in figure 11c is interesting. It appears there is a recirculation in here. Is there any explanation for this? Or do we need to reconsider the accuracy of the steady-state DPM tracking conducted here?

5. Grid independence testing (and sensitivity study of the DPM tracking) is still needed for this study.

The manuscript still needs to be further revised and checked in terms of its grammar and structure. Quick comments regarding these:

* research object should be research objectives. 

* Heading for section 4.3: analyze the flow field should be Flow field analysis

* Figures 1-4 can be combined together.

* 3rd sentence in the abstract: "After verifying the validity and appropriateness of the 11 simulation and the simulation found that the separation efficiency, static pressure distribution, and 12 total pressure loss generated as a result of the boundary conditions of the total pressure inlet and 13 flow outlet were close to the experimental values." --> This needs to be rephrased.
